# Emergence of Text Semantics in CLIP Image Encoders

**Sreeram Vennam** *
IIIT Hyderabad
sreeram.vennam@students.iiit.ac.in

**Shashwat Singh** *
IIIT Hyderabad
shashwat.s@research.iiit.ac.in

**Anirudh Govil**
IIIT Hyderabad
anirudhgovil2002@gmail.com

**Ponnurangam Kumaraguru**
IIIT Hyderabad
pk.guru@iiit.ac.in

## Abstract

Certain self-supervised approaches to train image encoders, like CLIP [13], align images with their text captions. However, these approaches do not have an *a priori* incentive to learn to associate text inside the image with the semantics of the text. Our work studies the semantics of text rendered in images. We show evidence suggesting that the image representations of CLIP have a subspace for textual semantics that abstracts away fonts. Furthermore, we show that the rendered text representations from the image encoder only slightly lag behind the text representations with respect to preserving semantic relationships.

## 1 Introduction

**Image Representations**    Self-supervised approaches have had remarkable success in textual tasks [11]. To harness the power of large-scale image data for dense image representations, [13] proposes a self supervised learning task that aligns images with textual descriptions of the image. Once trained, the image encoder has been shown to easily adapt to downstream image tasks.

**Processing text in image**    Processing text as an image mirrors nature; most humans consume text visually. Characterising how vision-language models process textual artifacts inside images is of great importance – for example, in the setting of sign boards for self driving cars. Processing rendered text has some relevance in the discussion of tokenization as well with [14, 16] discussing the bottlenecks current tokenization methods for language models for multi-lingual support and generalization.

However, it is not obvious that the semantics of an image should correspond to that of the text *inside* the image – the image of a large building with the sign board that says "Caesar's" corresponds to a hotel and not a historical figure. Especially in the context of the CLIP training task, the captions would typically be about the object that is in the image and not the text in the image. Nevertheless, preliminary evidences have been provided to show that the CLIP image encoders process text in some form with [10] experimentally demonstrating that CLIP's image representations can be used to match text inside an image with images of what the text describes. In contrast, our work studies the *textual semantics* of rendered text in the *absence of* a corresponding image. We deal with tasks like sentiment classification – for which no clear image grounding exists.

**Contributions**    We hypothesize that the Image encoder in CLIP captures the textual semantics of text rendered in images. Furthermore, we pose that the semantic representation is robust to strictly visual attributes like font. More specifically, our contributions are

---

*Equal Contribution

38th Conference on Neural Information Processing Systems (NeurIPS 2024).

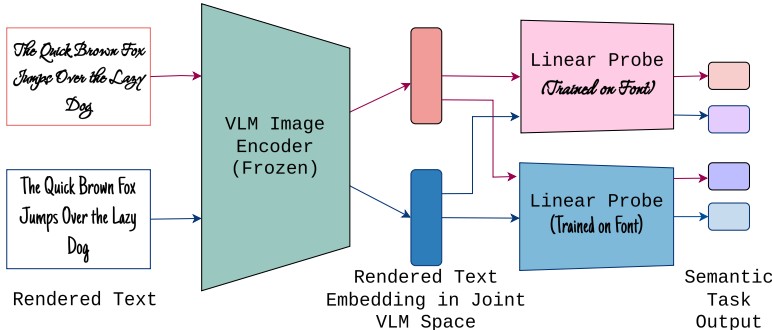

Figure 1: We train a linear probe per font and evaluate on every font. The task, sentiment classification, is purely linguistic and ideally is agnostic to the font used to render the text

- Using affine probes (see 2.1 for details) we provide evidence that decision boundaries can be drawn on image representations to classify sentiment. Furthermore, these boundaries generalize across the different fonts used to render the text.

- We provide evidence that rendered text representations from the CLIP image encoder is almost on par with the representations from its text encoder. As detailed in 2.2, we use second-order similarity alignment [8] with the representations from a pure Language Model Llama 3 [3] as a proxy for general expressiveness. Our methodology here derives inspiration from [6] and [7].

## 2   Experiments

**Encoding text using the image encoder**    To obtain representations of text from the CLIP image encoder, we render text into an image using [2]. We investigate textual semantics by checking for invariance to visual artifacts like fonts, and thus we render each text sample using multiple fonts. We use the pooled output of the CLIP image encoder as the 'representation' for the text.

### 2.1   Image Representations Affinely Encode Sentiment

We hypothesize that there exists a text-semantic subspace in the representation space of the image encoder. Following the linear representation hypothesis (LRH) [12], this implies that we can draw an affine decision boundary to classify linguistic attributes like sentiment. Furthermore, if there is a true text-semantic subspace in the image representations, then these decision boundaries should be robust to purely "visual" artifacts like fonts. Given a dataset $\mathcal{D} = \{(t_i, y_i)\}_{i=1}^{N}$ where the $t_i$ is the text and $y_i$ is the classification target, let $h_i^f$ be the image embedding of $t_i$ when rendered using font $f$. We then define an affine probe

$$y_i \sim \text{softmax}(W h_i^f + b).$$

We train one probe *per font* (see 1) on two established semantic classification datasets, SST2 [4] and MR [1]. Each probe is evaluated with a held out test set rendered using *every font* i.e. each probe is tested on every font; this tells us whether the learnt decision boundary is purely semantic and therefore agnostic to font. Furthermore, to provide a reasonable baseline, we construct a control task in line with [5] specified in 3.1.

### 2.2   Measuring Textual Semantics in Image Encodings

We formalize our study of the semantics encoded in the rendered image encoding *independent of a particular task*. We operationalize our measure of a notion of *general semantics* by using a second order isomorphism comparison with a fully pretrained Language Model Llama 3 [3]. The rationale for comparing these representations against LLama is that text-only models of its class offer the best

performance when it comes to capturing the semantics of text. For the actual comparison, we use the well known measure Central Kernel Alignment CKA [8] with a linear kernel.

Restating the linear CKA metric, for matrices of $n$ embeddings to be compared after centering $X, Y \in \mathbb{R}^{n \times d}$

$$\text{CKA}(X, Y) = \frac{\text{vec}(XX^T) \cdot \text{vec}(YY^T)}{||X^T X||_F^2 \, ||Y^T Y||_F^2}$$

Assuming that there are two sets of embeddings over the same set of samples, the CKA metric constructs a similarity matrix of each embedding with every other embedding in the set and then the similarity matrices of the two sets are compared. This intra-similarity measure allows us to compare embeddings from different kinds of models. Since this similarity number is arbitrary we contextualize the numbers obtained from the rendered text embedding by recognizing that they should be upper-bounded by the CKA score of the CLIP text encoder against Llama; we use the last hidden state from Llama as a sentence representation and the pooler embedding for the CLIP text encoder. Additionally for the lower bound, we use the controls as stated in 3. For this experiment, we use 1000 samples from the captions in the MS-COCO [9] dataset for the sentences. **

# 3 Controls

## 3.1 ROT-k

To root out the possibility of performance being obtained due to consistent character associations. We introduce a control that encrypts each sentence using the ROT-k cipher i.e. we substitute each letter with a letter $k = 9$ places later in the English Alphabet. The purpose of this control is to strip all the text of all semantics while preserving consistent character and keyword level patterns. This cipher provides a consistent mapping such that the transformed sentences are still self-consistent, i.e. if it possible to do the task by using token or character associations alone – it should be possible even after the cipher has been applied. This control therefore checks whether our tasks and results can be short-circuited by using character associations alone – if we are unable to train a probe as in 2.1 on top of these transformed input's representations then that implies that the task relies on actual semantics to be solved.

## 3.2 Jumble

To mitigate against inherent model bias to how characters look like, we introduce a control by randomizing the character sequence, including spaces before rendering the text. This produces an incomprehensible character sequence. This manipulation ensures that any meaningful semantic information is stripped. The model's performance should approximate random accuracy in this setting which validates the control. This control is run for all of our experiments.

# 4 Results

We run most of our experiments on two CLIP-based models OpenClip [13] and LAION-CLIP which is a CLIP based model trained on the LAION 5B dataset [15]. Our results are consistent across both the models. In 2 we see that the f1 scores by our sentiment probes are well above chance.

Furthermore, we observe that probes trained only on one font generalize directly to test sets of other fonts – providing us with the evidence for a semantic subspace that abstracts away the visual details of fonts. These fonts vary drastically in terms of visual style, Cedarville is handwritten, Source Code Pro is monospace, and Kenia is decorative with distinct gaps and curves. Results for all fonts are in the appendix A.1. In contrast, we see that the performance for the controls are consistently low – they are near 0.6 when tested on the same font and have no cross-font generalization (with the f1 being very close to random chance 0.5.) This gives us further confidence in the requirement of semantics to do well on the experimental setup of 2.1. We report results from the experiment in section 2.2 in Table 1 and we see that the CKA numbers of the image encoder only slightly lag behind that of the CLIP text encoder for most fonts (they are very close for popular fonts like Times New

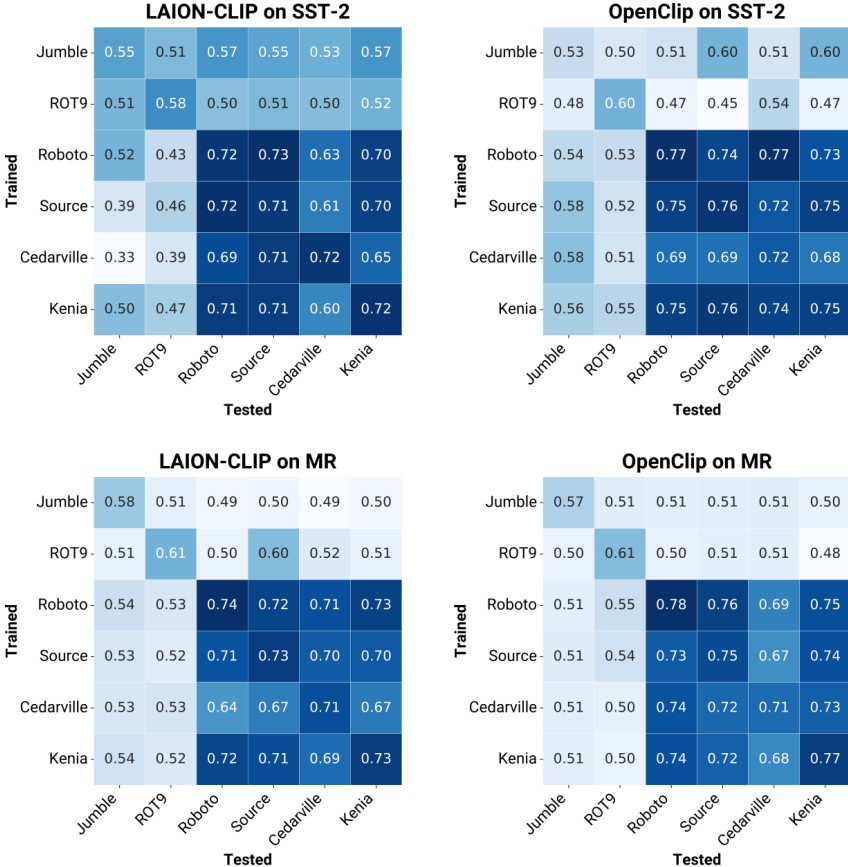

Figure 2: We report the f1-scores for binary classification. Each row represents the font on which the probe was trained and the column represents the test setting. We evalute them on two sentiment datasets SST-2 [4] and MR [1]

Roman). Note that Text-GT values are constant across fonts because there is no rendering involved. Moreover, the CKA numbers for the embeddings from the control settings are significantly worse than the meaningful font settings. These results provide evidence to the fact that the image encoder has non-trivial semantics encoded. Note that in both the cases, the controls Jumble and ROT-k are rendered in Roboto; we chose the font because it performs the best.

## 5  Conclusion

We conclude that the Contrastive Image-Language Pretraining over large parallel corpora leads to some textual semantics beinging capture by the image representations. This is supported by the fact that there is a semantic subspace in the image representations of rendered that is mostly agnostic to the font the text is rendered in. The control tests further provides us with an indication that rendered text is being encoded semantically and not via character associations. We believe this work should serve as motivation to studying the training or task circumstances that lead to the emergence of textual semantics in image encoders and perhaps better world understanding.

## 6  Acknowledgement

The authors of this paper would like to thank Abhinav S. Menon, Pratyaksh Gautam, Priyanshul Govil, and Shashwat Goel for their help and thoughtful comments.

Table 1: CKA Scores comparing the alignment between image representations of rendered-text (**Image**), text representations from the CLIP text encoder (**Text**) and the ground truth (**GT**) — LLama text representations on 1000 samples from MSCOCO [9].

| Experiment | OpenClip | | | LAION-CLIP | | |
|---|---|---|---|---|---|---|
| | **Image-GT** | **Text-GT** | **Image-Text** | **Image-GT** | **Text-GT** | **Image-Text** |
| Jumble | 0.04 | 0.03 | 0.10 | 0.06 | 0.06 | 0.38 |
| ROT13 | 0.07 | 0.10 | 0.17 | 0.06 | 0.17 | 0.35 |
| ROT9 | 0.05 | 0.11 | 0.13 | 0.07 | 0.19 | 0.34 |
| Roboto | 0.19 | 0.23 | 0.54 | 0.21 | 0.26 | 0.72 |
| Source | 0.19 | 0.23 | 0.59 | 0.19 | 0.26 | 0.66 |
| Times New Roman | 0.24 | 0.23 | 0.67 | 0.21 | 0.26 | 0.72 |
| Tiny5 | 0.24 | 0.23 | 0.61 | 0.20 | 0.26 | 0.65 |
| Just Another Hand | 0.16 | 0.23 | 0.41 | 0.16 | 0.26 | 0.51 |
| Pacifico | 0.24 | 0.23 | 0.67 | 0.21 | 0.26 | 0.72 |
| Cedarville | 0.18 | 0.23 | 0.52 | 0.18 | 0.26 | 0.63 |
| Grey Qo | 0.13 | 0.23 | 0.34 | 0.14 | 0.26 | 0.42 |
| Dancing Script | 0.22 | 0.23 | 0.65 | 0.20 | 0.26 | 0.69 |
| Shadows into Light | 0.23 | 0.23 | 0.66 | 0.20 | 0.26 | 0.70 |
| Playwright Peru | 0.21 | 0.23 | 0.62 | 0.19 | 0.26 | 0.68 |
| Kenia | 0.27 | 0.23 | 0.67 | 0.23 | 0.26 | 0.71 |

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

# A  Appendix

## A.1  Trained-on-Tested-on Plots

Results for all fonts in the sentiment experiment is present here.

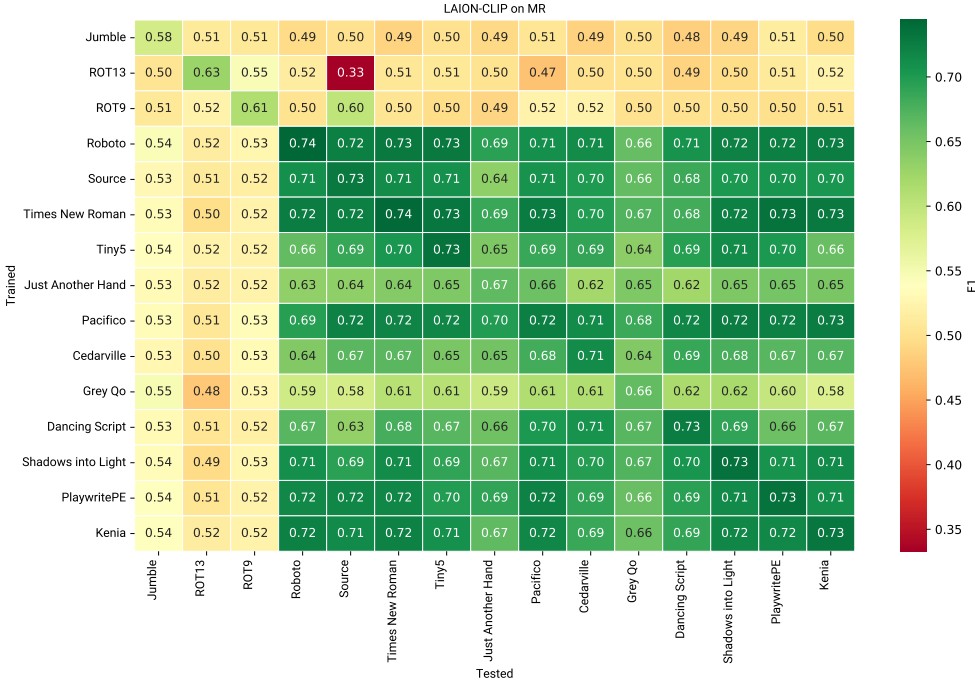

Figure 3: LAION-CLIP on MR

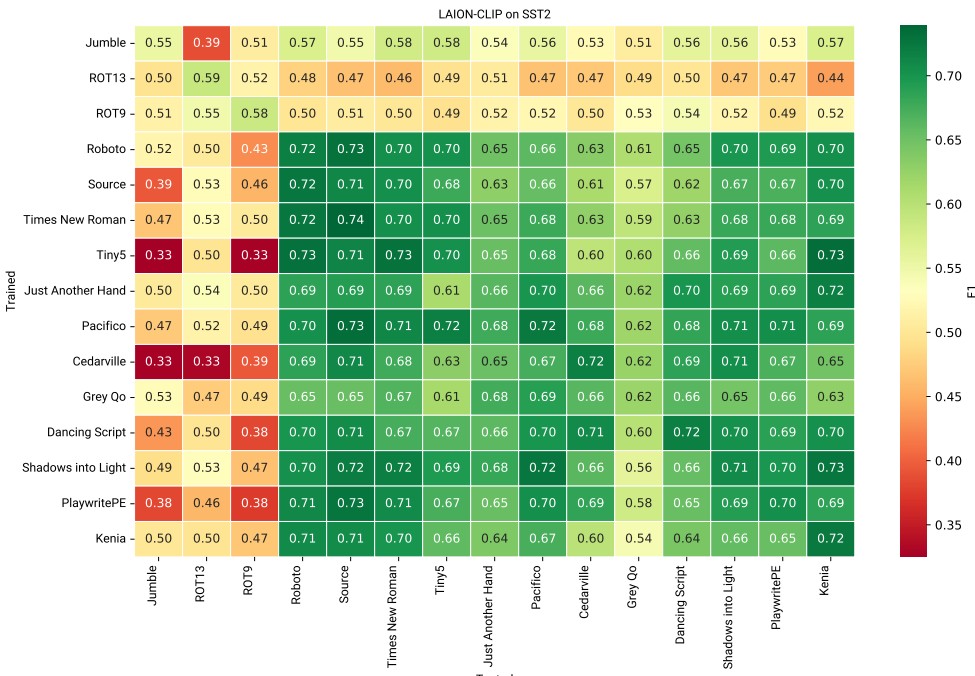

Figure 4: LAION-CLIP on SST2

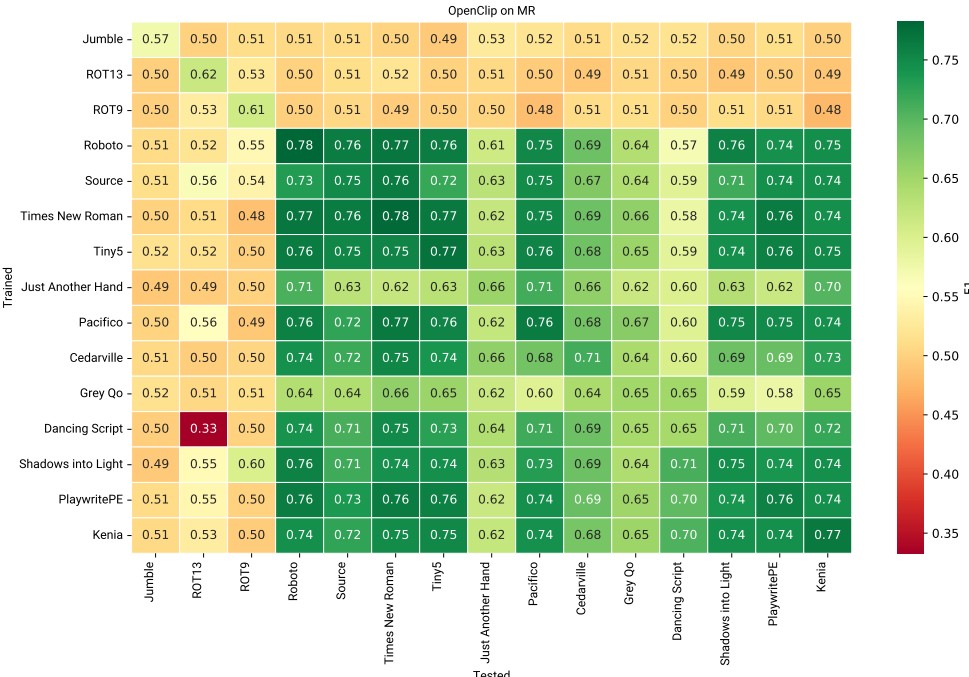

Figure 5: OpenClip on MR

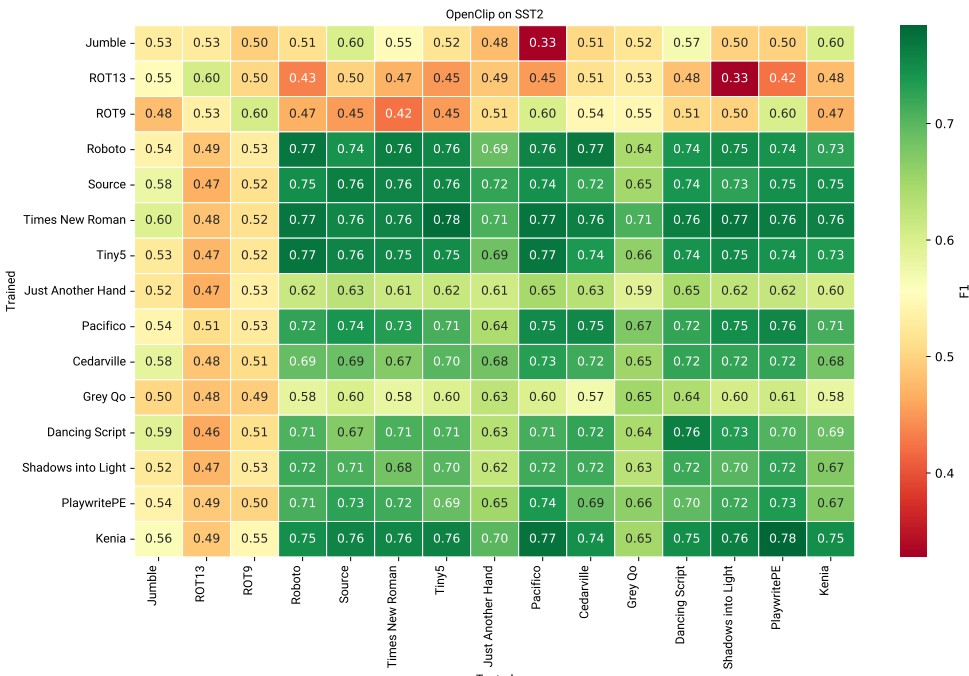

Figure 6: OpenClip on SST2

