# OpenReview forum: "Emergence of Text Semantics in CLIP Image Encoders"
_NeurIPS.cc/2024/Workshop/UniReps — UniReps_

### Official Review · Reviewer_uU5z · 2024-10-05
**interesting insight yet the paper is very incomplete**

**Rating:** 6
**Confidence:** 4

**Review:**

I. paper summary:

This paper brings an insight that CLIP has a subspace, which aligns the characters in the image with their actual semantic meanings.  The authors  conducted experiments on the sentiment classification task with a linear probe per font and evaluate on every font.  The font is rendered with different fonts to test CLIP's understanding of the texts in images.

II. pros:

1. The authors investigated the interesting question, whether CLIP understands texts rendered in images? This topic is related to multi-modality representation and thus is related to the topic of workshop.

2. The authors hypothesized that there exists a text-semantic subspace in the representation space of the image encoder and conduct experiments to prove it.


III. cons:

1. While it is an interesting topic to investigate whether CLIP understands texts in images, this paper is very incomplete. There are no practical images shown in the paper and we could not explicitly understand how solid is this conclusion.

2.  The authors utilize linear probing on sentiment classification with  synthetic training data rendered in  different fonts.  I wonder how CLIP image encoder itself performs on  this task?

3. Though this topic is of academic value, does it have any practical value in real world applications? Since  OCR techs have become very advanced nowadays. To understand the texts in images, we could either use OCR and then feed the characters to LLM or use VLLM to directly understand these texts. Especially for the VLLM approaches,   the accuracies are very high now. The author shall compare their methods with SOTA  VLLMs.

---

### Official Review · Reviewer_xSEF · 2024-10-05
**Interesting study on CILP image representations and possible encoding of textual semantics**

**Rating:** 7
**Confidence:** 5

**Review:**

The paper studies how CLIP image encoders capture the semantics of rendered text in images, beyond their original goal of aligning images with their text captions. The authors hypothesize that there is a semantic subspace in the image representation that can be used for text-based tasks like sentiment classification, independent of visual features such as font style. Through a series of experiments, they demonstrate that CLIP’s image representations can classify the sentiment of text and perform comparably to its text encoder, suggesting the emergence of non-trivial textual semantics. The use of controls, such as ROT-K cipher and jumbled text, seems to show that the model’s success is based on true semantic understanding rather than character associations.

Strengths:
- The topic is rather new and interesting.
- The tests are well performed: The authors use a set of controls, such as the ROT-K and Jumble tests, to ensure that the model’s performance is genuinely due to semantic understanding rather than superficial character associations or visual cues; and the experiments are performed on multiple CLIP-based models (OpenCLIP and LAION-CLIP).

Weaknesses:
- The use of LLaMA and dependence on Central Kernel Alignment (CKA) might not fully represent the semantic structure emerging in image encoders, as these comparisons could introduce biases from the pretrained text-only models.
- Even if the study includes controls like Jumble and ROT-K, it lacks a thorough breakdown of why performance varies under these settings.

---

### Official Review · Reviewer_9zon · 2024-10-07
**Paper with opportunities for further development**

**Rating:** 6
**Confidence:** 4

**Review:**

### Summary
- This paper explores the concept that the image encoder of a Vision-Language Model (VLM) can capture text semantics within images. Using the LRH approach, it trains an affine probe for each font to ensure generalization across different fonts, demonstrating its effectiveness through F1 scores from a text semantic classification task. Additionally, the similarity comparison between CLIP image and text encodings is presented to support the claim that the image encoder can indeed capture text semantics.

### Strength

- The concept that an image encoder can capture text semantics is interesting.

### Weakness

- The introduction of this paper lacks fluency and could benefit from a more organized flow. For instance, the mention of tokenization methods between lines 15 and 17 appears unnecessary.
- While the paper claims that image-GT is comparable to text-GT as evidence for the image encoder's validity, the similarity between CLIP image and text representations seems relatively low. If the similarity is not actually low, it would be beneficial to present baseline comparisons to demonstrate this. However, if the similarity is indeed low, this discrepancy raises important questions about the encoder’s validity, and it would be helpful to provide an explanation for this.

---

### Decision · Program_Chairs · 2024-10-10

**Decision:**

Accept

**Comment:**

In light of the positive reviewers' feedback and relevancy of the submission, we are pleased to accept this paper for presentation at UniReps 2024. We kindly ask the authors to incorporate the reviewers' suggestions and feedback in the final camera-ready version of the manuscript.